# *Lactobacillus delbrueckii* subsp. *bulgaricus* 2038 and *Streptococcus thermophilus* 1131 Induce the Expression of the REG3 Family in the Small Intestine of Mice via the Stimulation of Dendritic Cells and Type 3 Innate Lymphoid Cells

**DOI:** 10.3390/nu11122998

**Published:** 2019-12-07

**Authors:** Kyosuke Kobayashi, Yoshiko Honme, Toshihiro Sashihara

**Affiliations:** 1Applied Microbiology Research Department, Food Microbiology and Function Research Laboratories, R&D Division, Meiji Co., Ltd., Hachiouji, Tokyo 192-0919, Japan; toshihiro.sashihara@meiji.com; 2Basic Microbiology Research Department, Food Microbiology and Function Research Laboratories, R&D Division, Meiji Co., Ltd., Hachiouji, Tokyo 192-0919, Japan; yoshiko.honme@meiji.com

**Keywords:** lactic acid bacteria, dendritic cell, type 3 innate immune cell, interleukin-23, interleukin-22, REG3 family

## Abstract

Accumulating evidence clarifies that intestinal barrier function, for example, by the mucus layer, antimicrobial peptides, immune systems, and epithelial tight junctions, plays crucial roles in maintaining our health. We reported previously that yogurt fermented with *Lactobacillus delbrueckii* subsp. *bulgaricus* 2038 and *Streptococcus thermophilus* 1131 induced the gene expression of the regenerating family member 3 (REG3) family, which encodes antimicrobial peptides in the small intestine, although it was unclear how the yogurt activated the intestinal cells related to it. Here, we evaluated the cytokine production from the intestinal immune cells stimulated by these strains in vitro and in vivo to elucidate the mechanism for the induction of the REG3 family by the yogurt. The results showed that stimulation by both strains induced interleukin (IL)-23 production from bone marrow-derived dendritic cells (DCs) and IL-22 production from small intestinal lamina propria (LP) cells. In addition, oral administration of these strains to mice increased IL-23p19^+^ LPDCs and IL-22^+^ type 3 innate lymphoid cells and induced the expression of *Reg3g* in small intestinal tissue. Moreover, we showed that the activities for the induction of IL-23 by DCs were strain dependent on *L. bulgaricus* and *S. thermophilus* and that *S. thermophilus* 1131, which is the predominant species in the yogurt, exhibited relatively higher activity compared to the other strains of *S. thermophilus*. Our findings suggested that these yogurt starter strains, *L. bulgaricus* 2038 and *S. thermophilus* 1131, have the potential to maintain and improve intestinal barrier function by stimulating immune cells in the LP.

## 1. Introduction

The intestines, with a huge surface area, have the necessary function to absorb molecules such as water and nutrients to maintain homeostasis. However, they are exposed to a great number of pathogens and toxins and, therefore, have a barrier function to appropriately eliminate such unnecessary molecules [1,2]. The reduction of intestinal barrier function is associated with the development of diseases such as inflammatory bowel disease [3], and, more importantly, it is reported to cause the dysfunction of various tissues [4]. Therefore, to maintain or strengthen intestinal barrier function is important for health.

Lactic acid bacteria (LAB) have been widely used as a starter for the fermentation of various foods and ingested all over the world since ancient times. They have been reported to increase the number of defecations [5], to stimulate immune activity [6], and to ameliorate the symptoms of inflammatory bowel disease [7], for example. In addition, we recently reported that the gene expression of the regenerating family member 3 (REG3) family, which encodes antimicrobial peptides that function to maintain the intestinal barrier, was reduced in the small intestines of mice due to aging, whereas the long-term ingestion of yogurt fermented with *Lactobacillus delbrueckii* subsp. *bulgaricus* 2038 and *Streptococcus thermophilus* 1131 by aged mice induced its expression [8].

The REG3 family is expressed in the small intestinal epithelial cells [9,10]. It has been reported that REG3β exerts bactericidal activity against Gram-negative bacteria (e.g., *Salmonella enteritidis* [11] and *Yersinia pseudotuberculosis* [12]) and some Gram-positive bacteria [13], whereas REG3γ exerts bactericidal activity against Gram-positive bacteria (e.g., *Listeria monocytogenes* [14]). Hence, the REG3 family is suggested to contribute to the suppression of infections via the small intestine. In addition, REG3γ is essential to physically separate the microbiota from the small intestinal epithelial surface and limits the excess activation of immune responses by them [15]. Furthermore, the increase in intestinal REG3γ suppresses the apoptosis of stem cells and Paneth cells induced by graft-versus-host disease, suggesting that REG3γ has an important role in small intestinal barrier function [16].

Here, we analyzed the mechanism by which yogurt induced the gene expression of the REG3 family that has the several aforementioned functions. Because germ-free mice and mice administered with antibiotics showed a low expression of the REG3 family genes [17], it is suggested that starter LAB, but not the nutrients and ingredients in the yogurt, have the potential to induce genes of the REG3 family. Besides this, the REG3 family is induced in the intestinal epithelial cells by recognizing interleukin (IL)-22 [18], which is predominantly produced in the small intestine by type 3 innate lymphoid cell (ILC3) [19]. In addition, IL-23, a potent inducer of IL-22 [20], is produced by dendritic cells (DCs) and macrophages and has been reported to be induced by the stimulation of the bacterial cell component via Toll-like receptors (TLRs) on immune cells [20]. Therefore, we hypothesized that either *L. bulgaricus* 2038 or *S. thermophilus* 1131 or both induced the gene expression of the REG3 family by stimulating the production of these cytokines in immune cells in the small intestinal lamina propria (LP).

In this study, we investigated the activities of *L. bulgaricus* 2038 and *S. thermophilus* 1131 for the stimulation of cytokine production related to expression of the REG3 family in vitro using immune cells from mice. Then, we further verified whether these reactions were observed in the small intestine when orally administered in vivo.

## 2. Materials and Methods 

### 2.1. Mice

Ten-week-old specific pathogen-free (SPF) male BALB/c mice were purchased from Charles River Laboratories Japan (Kanagawa, Japan) and used for the preparation of bone marrow-derived DCs (BMDCs). Eight-week-old SPF male BALB/c mice were purchased from Japan SLC (Shizuoka, Japan) and used for the preparation of small intestinal lamina propria lymphocytes (LPLs). For experiments to isolate immune cells, mice were housed with group breeding and fed CRF-1 (Oriental Yeast, Tokyo, Japan), a purified diet. Six-month-old SPF male ICR mice were purchased from Japan SLC and used for LAB administration experiments. For these experiments, mice were housed individually and fed MF (Oriental Yeast, Tokyo, Japan), a non-purified diet. For all experiments, cervical dislocation was conducted for euthanasia.

All study protocols were approved by the Animal Experimental Committee of Meiji Co., Ltd. (approval no. 2016_3871_0068, approval date June 6, 2016; approval no. 2017_3871_0032, approval date April 26, 2017; approval no. 2017_3871_0039, approval date April 28, 2017 and approval no. 2017_3871_0182, approval date October 3, 2017). The experiments were performed according to the guidelines of the committee.

### 2.2. LAB Culture

*L. bulgaricus* and *S. thermophilus* were cultured anaerobically in de Man Rogosa Sharpe broth (Becton Dickinson, Cockeysville, MD, USA) and M17 broth (Becton Dickinson, Cockeysville, MD, USA) supplemented with 1% lactose, respectively, at 37 °C for 18 h. For in vitro cytokine production assays, bacteria were washed twice with phosphate-buffered saline (PBS; pH 7.4) and suspended in PBS so that the optical density at 600 nm was 2.0 using U-2810 spectrophotometer (Hitachi, Tokyo, Japan). Then, bacteria were heat-killed at 75 °C for 1 h.

All bacteria used in this study are the property of Meiji Co., Ltd.

### 2.3. In vitro Experiment of IL-23 Production Using DCs

BMDCs were generated from cells collected from the tibia and femur of BALB/c mice as described previously [21]. After red blood cell lysis and depletion of cells expressing cluster of differentiation (CD) 4, CD8, and major histocompatibility complex (MHC) II (I-A/I-E) using magnetic beads (Miltenyi Biotec, Bergisch Gladbach, Germany) and autoMACS (Miltenyi Biotec, Bergisch Gladbach, Germany) for excluding T cells and antigen-presenting cells, cells were cultured in Roswell Park Memorial Institute medium 1640 (RPMI 1640; Invitrogen, Carlsbad, CA, USA) containing 10% fetal bovine serum (FBS) and 10% granulocyte-macrophage colony-stimulating factor. On day 8, non-adherent and loosely adherent cells were harvested.

BMDCs were cultured in RPMI 1640 containing 10% FBS, 100 U/mL penicillin, and 100 µg/mL streptomycin at a concentration of 1.0 × 10^6^ cells/mL in a 24-well culture plate (0.5 mL/well) at 37 °C in 5% CO_2_. Five microliters of PBS or heat-killed *L. bulgaricus* 2038 or *S. thermophilus* 1131 were added to the medium and incubated for 24 h. Culture supernatants were collected for the determination of IL-23 levels.

Temperature-sensitive DCs (tsDCs), an immortalized mouse DC line that was established from the bone marrow [22], were purchased and directly obtained from the European Collection of Authenticated Cell Cultures (Salisbury, UK; catalog no. 01081609). tsDCs were cultured in Iscove’s MEM GlutaMAX Supplement (Invitrogen, Carlsbad, CA, USA) containing 5% FBS, 2 mM glutamine, 100 U/mL penicillin, 100 μg/mL streptomycin, and 0.1% 2-mercaptoethanol at 5.0 × 10^5^ cells/mL in a 96-well culture plate (0.1 mL/well) at 33 °C in 9% CO_2_. One microliter of PBS or heat-killed strains of *L. bulgaricus* and *S. thermophilus* was added to the medium and incubated for 24 h. Culture supernatants were collected for the determination of IL-23 levels.

### 2.4. In vitro Experiment of IL-22 Production Using Intestinal LPLs

For small intestinal LPL preparations, the small intestines were extracted, the attached fat was removed, and tissues were cut open longitudinally. Luminal contents were removed by flushing with cold PBS. The small intestines were minced to small pieces and washed thrice with FACS Flow (Becton Dickinson) containing 0.5% bovine serum albumin (BSA; Wako, Osaka, Japan). Epithelial cells and intraepithelial lymphocytes were removed by shaking tissue in buffer [Hank’s balanced salt solution (HBSS)(-); Life Technologies, Carlsbad, CA, USA] containing 10 mM 4-(2-hydroxyethyl)-1-piperazineethanesulfonic acid (HEPES; Life Technologies, Carlsbad, CA, USA), 5 mM EDTA (Wako, Osaka, Japan), 1 mM dithiothreitol (Wako, Osaka, Japan) and 1% BSA for 30 min at 37 °C with MACS tube rotator (Miltenyi Biotec, Bergisch Gladbach, Germany). Then, samples were washed thrice with FACS Flow (Becton Dickinson, Cockeysville, MD, USA) containing 0.5% BSA and twice with HBSS(+) containing 10 mM HEPES and 1% BSA. The LP layer was isolated by digesting the remaining tissue with a Lamina Propria Dissociation Kit (Miltenyi Biotec, Bergisch Gladbach, Germany) according to the manufacturer’s protocol. Cells were filtered with a 100 μm cell strainer (Becton Dickinson, Cockeysville, MD, USA) and centrifuged at 1000 rpm for 5 min. To obtain purified LPLs, pellets were suspended with 2 mL of 100% Percoll (GE Healthcare, Buckinghamshire, UK) and overlaid with 2 mL of pre-chilled 40% Percoll carefully as not to disturb the phases and centrifuged at 1500 rpm for 20 min; the inter-phase ring was transferred to a new tube. Cell numbers were counted using Nucleo Counter NC-100 (Chemometec, Copenhagen, Denmark).

LPLs were cultured in the same conditions as BMDCs, except that the cultivation time was 48 h, to determine the IL-22 levels in the supernatant.

### 2.5. Administration of LAB

LAB were cultured according to “LAB culture” above. Then, they were washed twice with PBS and adjusted to 5 × 10^8^ colony forming unit (CFU)/mL.

Live bacteria were orally gavaged for eight consecutive days at dose of 0.2 mL/mouse. Control mice were given an equal volume of PBS.

Mice were sacrificed by cervical dislocation 2 or 3 h after the last administration. The small intestines were extracted, and 1 cm of the jejunum was collected and kept at 4 °C in RNA later (Invitrogen, Carlsbad, CA, USA). Other small intestines after the removal of Peyer’s patches were used for LPL isolation.

### 2.6. Enzyme-Linked Immunosorbent Assay (ELISA)

IL-23 concentrations in BMDC supernatants and IL-22 concentrations in LPL supernatants were determined using Mouse IL-23 Quantikine ELISA Kit (R&D Systems, Minneapolis, MN, USA) and Mouse/Rat IL-22 Quantikine ELISA Kit (R&D Systems, Minneapolis, MN, USA), respectively.

### 2.7. Flow Cytometry

LPLs isolated from the small intestines from the LAB administration experiment were plated at 3.0 × 10^6^ cells/mL in a 96-well plate (0.1 mL/well). Before staining, Fc receptors were blocked with purified anti-CD16/CD32 (2.4G2) for 10 min at 4 °C. LPLs were surface stained with fluorescein isothiocyanate-anti-Lin cocktail (anti-CD3, 17A2; anti-CD45R, RA3-6B2; anti-CD11b, M1/70; anti-TER-119, TER-119; and anti-Ly-G6, RB6-8C5), Allophycocyanin (APC)-anti-CD11c (N418), APC-H7-anti-CD117 (2B8), and BV421-anti-CD127 (SB/199) mixed with Fixable Viability Dye eFluor 506 to stain dead cells. Cells were fixed and permeabilized with 1× Fix/Perm Buffer in Transcription Factor Buffer Set (BD Biosciences, San Jose, CA, USA) and then washed twice with 1× Perm/Wash Buffer in the same kit. For intracellular staining, cells were stained with phycoerythrin (PE)-anti-retinoic acid receptor-related orphan receptor gamma t (RORγt) (AFKJS), Alexa Fluor 488-anti-IL-23p19 (fc23cpg), and Alexa Fluor 647-anti-IL-22 (Poly5164). Data were acquired using FACSVerse flow cytometer (BD Biosciences, San Jose, CA). CD11c^high^ cells were analyzed as LPDCs. Lin (CD3, CD45R, CD11b, TER119, Ly-G6)^-^ CD117^+^ CD127^+^ RORγt^+^ cells were analyzed as ILC3s.

### 2.8. RNA Isolation and Gene Expression Analysis

Total RNA was prepared from the jejunum using NucleoSpin RNA (Macherey-Nagel, Düren, Germany) according to the manufacturer’s protocols. RNA was quantified and assessed for purity using 2100 Bioanalyzer System (Agilent, Santa Clara, CA, USA). Complementary DNA was synthesized from 1 μg total RNA using PrimeScript RT Master Mix (TaKaRa Bio, Shiga, Japan), and real-time polymerase chain reaction (PCR) was performed using a TaKaRa PCR Thermal Cycler Dice (TaKaRa Bio, Shiga, Japan) and SYBR Premix Ex Taq II (TaKaRa Bio, Shiga, Japan) according to the manufacturer’s protocol. Primers used were *Reg3g* forward 5′-TCAGGACATCTTGTGTCTGTGCTC-3′ and reverse 5′-CATCCACCTCTGTTGGGTTCA-3′ and *Gapdh* forward 5′-AAATGGTGAAGGTCGGTGTG-3′ and reverse 5′-TGAAGGGGTCGTTGATGG-3′. Quantitative comparisons were obtained using the ΔΔC_T_ method. ΔC_T_ was calculated as the value of C_T_ of *Reg3g* minus the C_T_ of *Gapdh*. ΔΔC_T_ was calculated as the value of ΔC_T_ of the treated group minus the ΔC_T_ of the control group. Thus, 2^−ΔΔCT^ was the normalized relative expression value in each group.

### 2.9. Statistics

Data are presented as the mean ± standard error (SE). The Shapiro–Wilk test was performed to determine the normality of distribution. If the data followed normal distribution, Dunnett’s test was used—if not, Steel’s test was performed. *p* < 0.05 was considered significant.

## 3. Results

### 3.1. In vitro Stimulation of Immune Cells with *L. bulgaricus* 2038 and *S. thermophilus* 1131 Led to Cytokine Production

We examined whether BMDCs, as a model of DCs in the small intestine, produced IL-23 by recognizing *L. bulgaricus* 2038 and *S. thermophilus* 1131 in vitro. It was assumed that live bacterial cells increase during co-cultivation with BMDCs and play probiotic activity roles such as colonization to BMDCs; therefore, heat-killed bacterial cells were used in in vitro experiments. Both heat-killed *L. bulgaricus* 2038 and *S. thermophilus* 1131 significantly induced IL-23 secretion into the medium (Figure 1a). It was reported that IL-23 stimulated IL-22 production from ILC3s in the LP [20]. We, therefore, estimated that these strains induced IL-22 production from LPLs isolated from the small intestinal LP. The results showed that these strains significantly induced IL-22 secretion into the medium as well (Figure 1b). *S. thermophilus* 1131 had higher activities compared to *L. bulgaricus* 2038 as observed in IL-23 production by BMDCs.

### 3.2. Oral Administration of *L. bulgaricus* 2038 and *S. thermophilus* 1131 Induced the Gene Expression of the REG3 Family in the Small Intestine

*L. bulgaricus* 2038 and *S. thermophilus* 1131 significantly increased IL-22 production from LPLs in vitro; hence, we estimated that the administration of these strains induced the gene expression of the REG3 family on the small intestinal epithelial cells by stimulating immune cells in LPLs. To test this, we analyzed the gene expression of the REG3 family in the small intestine from mice administered with these strains for eight consecutive days by real-time PCR. The expression of *Reg3g* in the jejunum was significantly induced by oral administration of *S. thermophilus* 1131 (Figure 2).

### 3.3. Oral Administration of *L. bulgaricus* 2038 and *S. thermophilus* 1131 Induced IL-23 Production from LPDCs and IL-22 Production from ILC3s

Next, to verify whether IL-23 production from DCs and IL-22 production from ILC3s in the LP were induced by *L. bulgaricus* 2038 and *S. thermophilus* 1131 administration as expected, we analyzed the intracellular cytokine production of LPLs isolated from mice administered with these strains for eight consecutive days using flow cytometry.

IL-23 is a heterodimeric cytokine composed of a p40 subunit shared with IL-12 and a unique p19 subunit [23]. Therefore, we analyzed IL-23p19^+^ cells as IL-23^+^ cells. The gating strategy for LPDCs is shown in Figure 3a. The rate of IL-23p19^+^ LPDCs per total LPDCs of *L. bulgaricus* 2038 and *S. thermophilus* 1131 groups was significantly higher than that of the control group (Figure 3b). Furthermore, we assessed whether the administration of these strains resulted in IL-22 production from ILC3s. We found that the rate of IL-22^+^ ILC3s per total ILC3s of the *S. thermophilus* 1131 group was significantly higher than that of the control group (Figure 4).

Overall, the intensity to induce both IL-23p19^+^ LPDCs and IL-22^+^ ILC3 by these strains in vivo corresponded to that to induce the expression of *Reg3g* and cytokine production in vitro.

### 3.4. Stimulation of IL-23 Production Was Strain dependent on *L. bulgaricus* and *S. thermophilus*

To assess whether the stimulation of cytokine production from LPLs was specific for *L. bulgaricus* 2038 and *S. thermophilus* 1131 or a common feature in *L. bulgaricus* and *S. thermophilus*, we analyzed the activity of other strains in these species to induce IL-23 production by tsDC. The results showed that the activities in both *L. bulgaricus* and *S. thermophilus* were strain-dependent and that several strains had no activity at all (Figure 5). Many *L. bulgaricus* strains showed higher activity than *L. bulgaricus* 2038, whereas *S. thermophilus* 1131 showed relatively high activity compared to the other strains of *S. thermophilus*.

## 4. Discussion

In the present report, we elucidated whether oral administration of yogurt starter strains, *L. bulgaricus* 2038 and *S. thermophilus* 1131, induced the gene expression of the REG3 family in mice small intestines via the stimulation of cytokine production in LPLs. A similar observation was reported regarding the induction of the gene expression of the REG3 family by LAB in the small intestine [24]. Hou et al. showed in the experiment that the expression in mice intestinal organoids was induced by co-cultivation with LPLs isolated from the small intestinal LP and *Lactobacillus reuteri* D8 [24]. These results suggested that *L. reuteri* D8 stimulated any immune cells in LPLs and consequently induced the expression of the REG3 family. However, it remains unclear which cells in LPLs contributed to the induction.

We clarified the activities of *L. bulgaricus* 2038 and *S. thermophilus* 1131 for IL-23 and IL-22 production from BMDCs and LPLs, respectively, in vitro. Moreover, oral administration of these strains into mice showed *Reg3g* induction in the small intestinal cells and especially elucidated IL-23 and IL-22 production from LPDC and ILC3, respectively, in vivo. Interestingly, the activities of *S. thermophilus* 1131 were higher compared to *L. bulgaricus* 2038 in all experiments. In our previous report, which showed that yogurt administration induced the gene expression of the REG3 family in the small intestine [8], yogurt fed to mice contained 10^7^ cells of *L. bulgaricus* 2038 and 10^8^ cells of *S. thermophilus* 1131 per gram. Considering the data in the present study, it is suggested that the induction of the gene expression of the REG3 family by yogurt in the previous report was mainly attributed to the activity of *S. thermophilus* 1131.

These strains first contact LPDCs in our hypothesis; hence, it is presumed that the activities of IL-23 production from LPDCs are correlated with those of IL-22 production from ILC3s and induction of REG3 genes from the small intestinal epithelial cells. Molecules recognized by TLRs and Nod-like receptors [25] expressed on the cell surfaces and intracellularly induce IL-23 production from DCs; therefore, various bacterial cell components derived from these strains influence the differences of the activities between them. Moreover, we showed that the activities of IL-23 production from DCs were strain dependent on *L. bulgaricus* and *S. thermophilus*. It is, therefore, possible to screen the strains that have high activities of IL-22 production and induction of REG3 gene expression.

In addition, for the route of *L. bulgaricus* 2038 and *S. thermophilus* 1131 to be recognized by DCs, it is deduced that they are transcytosed to the small intestinal LP via M cells, present on the follicle-associated epithelium (FAE), which have the role to transport luminal organisms passing through the LP [26]. It has been reported that the mucin layer is thinner in the FAE than villous because goblet cell differentiation is diminished by Notch signal from stromal cells underneath the FAE [27,28]. Therefore, it is estimated that administered *L. bulgaricus* 2038 and *S. thermophilus* 1131 readily access M cells and are transported. It has actually confirmed that *Lactobacillus salivarius* translocated across M cells in mice [29].

However, the limitations of this study should be noted. First, although not only DCs but also macrophages produce IL-23 [20], we have not yet analyzed this possibility in vivo. Because we have already clarified that *L. bulgaricus* 2038 and *S. thermophilus* 1131 induced IL-23 production from mouse peritoneal macrophages (Appendix A), it is estimated that these strains have the potential to activate IL-23 production from macrophages in the LP. Second, the causal relationship among the production of IL-23, IL-22 and the induction of REG3 gene expression could not be clarified in this experimental design. It has been reported that RORγt^+^ ILCs mainly product IL-22 in the small intestinal LP and mice deficient in RORγt had significant low gene expression of the REG3 family [30]. Therefore, it is suggested that the induction of IL-23 and IL-22 by bacteria mainly leads to the induction of *Reg3g*.

In this study, we clarified that yogurt starter strains, *L. bulgaricus* 2038 and *S. thermophilus* 1131, induced the expression of *Reg3g*. In addition, the expression was found to be associated with IL-23 and IL-22 production from LPDCs and ILC3s in the LP, respectively. Our present data suggested that these strains have the potential to improve intestinal barrier function, protecting from pathogens, maintaining a physical distance from the microbiota and the host mucosal surface, and suppressing the apoptosis of intestinal stem cells via the stimulation of these cells. We will further progress the research on how *L. bulgaricus* 2038 and *S. thermophilus* 1131 intake affects the phenotype with regard to intestinal barrier function.

## Figures and Tables

**Figure 1 nutrients-11-02998-f001:**
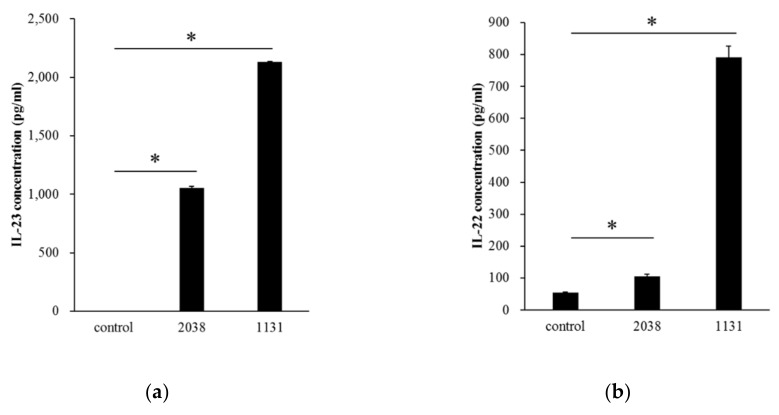
*Lactobacillus*
*bulgaricus* 2038 and *Streptococcus thermophilus* 1131 induced cytokine production from bone marrow-derived dendritic cells (BMDCs) and lamina propria lymphocytes (LPLs). (**a**) BMDCs were cultivated in the presence of heat-killed *L. bulgaricus* 2038 and *S. thermophilus* 1131 for 24 h, and supernatants were determined for interleukin (IL)-23 by enzyme-linked immunosorbent assay (ELISA) (*n* = 3 per group). (**b**) LPLs were cultivated in the same manner as BMDC but for 48 h, and supernatants were determined for IL-22 by ELISA (*n* = 3 per group). Data are the mean ± SE. * *p* < 0.05 (Dunnett’s test).

**Figure 2 nutrients-11-02998-f002:**
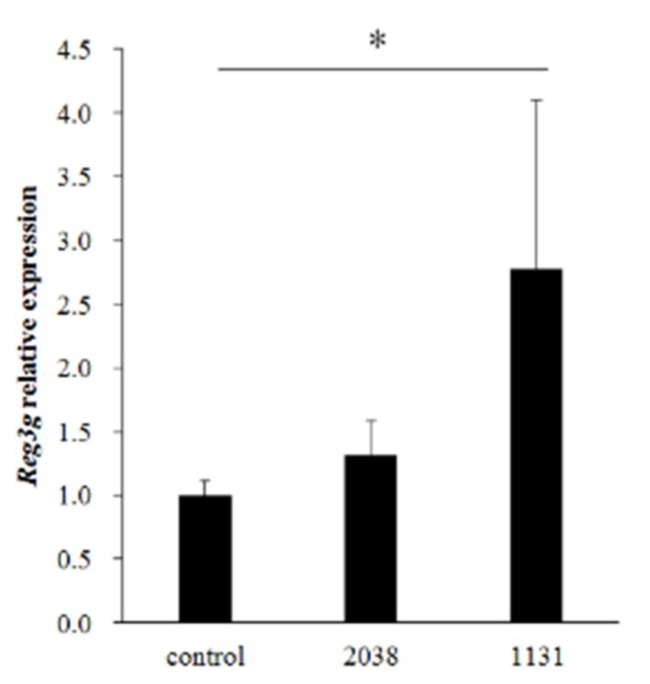
*L. bulgaricus* 2038 and *S. thermophilus* 1131 induced the expression of *Reg3g* in the jejunum. *L. bulgaricus* 2038 and *S. thermophilus* 1131 were orally administered to mice for eight consecutive days at a dose of 1 × 10^8^ CFU/mouse (*n* = 8 per group). Control group mice were given an equal volume of PBS (*n* = 8). Mice were sacrificed 3 h after the last administration. Changes in *Reg3g* expression levels were analyzed by real-time PCR. Data were normalized with *Gapdh* expression and showed relative expression level. Data are the mean ± SE. * *p* < 0.05 (Steel’s test).

**Figure 3 nutrients-11-02998-f003:**
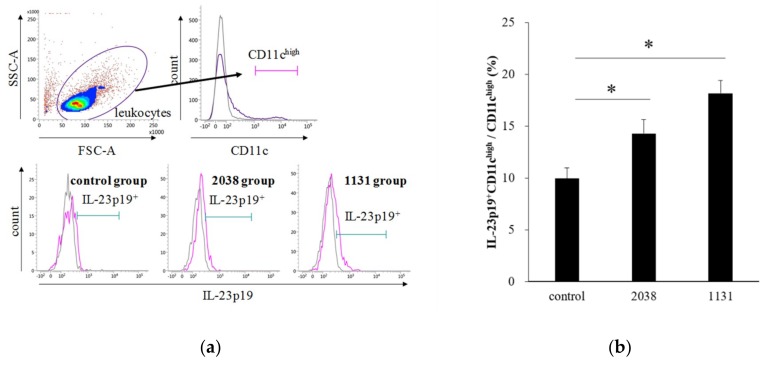
Oral administration of *L. bulgaricus* 2038 and *S. thermophilus* 1131 induced IL-23p19 production from LPDCs. *L. bulgaricus* 2038 and *S. thermophilus* 1131 were orally administered to mice for eight consecutive days at a dose of 1 × 10^8^ CFU/mouse (*n* = 10 per group). Control group mice were given an equal volume of PBS (*n* = 10). Mice were sacrificed 2 h after the last administration. (**a**) Gating strategy for LPDCs. Histograms for IL-23p19^+^ cells (purple line) of experimental groups are shown at the bottom (gray line; isotype control). (**b**) Rate of IL-23p19^+^ LPDCs per total LPDCs. Data are the mean ± SE. * *p* < 0.05 (Dunnett’s test).

**Figure 4 nutrients-11-02998-f004:**
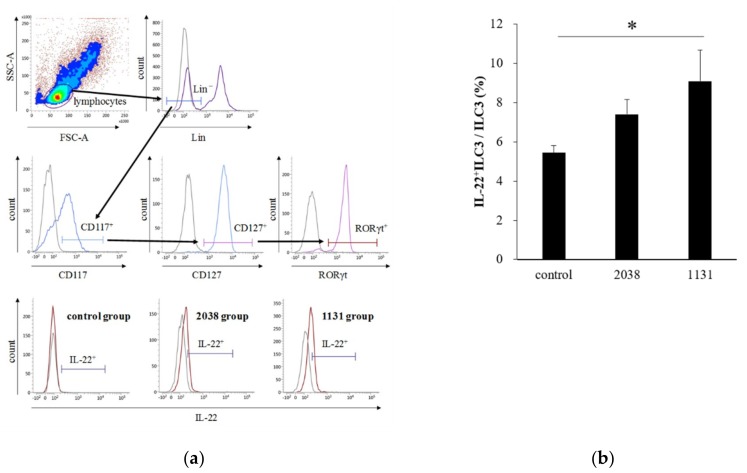
Oral administration of *L. bulgaricus* 2038 and *S. thermophilus* 1131 induced IL-22 production from ILC3s. *L. bulgaricus* 2038 and *S. thermophilus* 1131 were orally administered to mice daily for eight consecutive days at a dose of 1 × 10^8^ CFU/mouse (*n* = 8 per group). Control group mice were given an equal volume of PBS (*n* = 8). Mice were sacrificed 3 h after the last administration. (**a**) Gating strategy for IL-22^+^ ILC3s. Histograms for IL-22^+^ cells (brown line) of experimental groups are shown at the bottom (gray line; isotype control). (**b**) Rate of IL-22^+^ ILC3s per total ILC3s. Data are the mean ± SE. * *p* < 0.05 (Dunnett’s test).

**Figure 5 nutrients-11-02998-f005:**
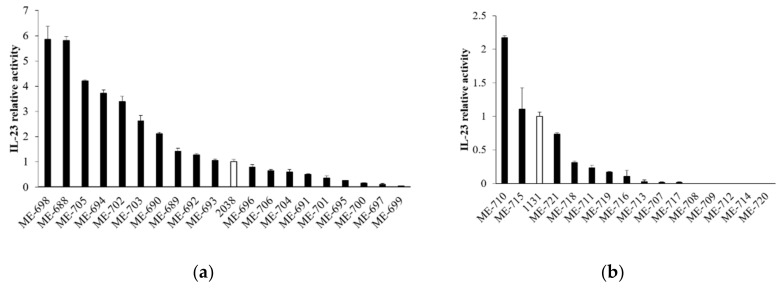
*L. bulgaricus* and *S. thermophilus* strains induced IL-23 production from tsDC in various degrees. tsDCs were cultivated in the presence of heat-killed (**a**) *L. bulgaricus* and (**b**) *S. thermophilus* for 24 h, and supernatants were determined for IL-23 by ELISA (*n* = 3 per group). White bars show *L. bulgaricus* 2038 or *S. thermophilus* 1131. Relative IL-23 activities were calculated as the quotient of the average IL-23 production by each *L. bulgaricus* or *S. thermophilus* divided by the average IL-23 production by *L. bulgaricus* 2038 or *S. thermophilus* 1131, respectively. Data are the mean ± SE.

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
