# Peer review of "Lactobacillus delbrueckii subsp. bulgaricus 2038 and Streptococcus thermophilus 1131 Induce the Expression of the REG3 Family in the Small Intestine of Mice via the Stimulation of Dendritic Cells and Type 3 Innate Lymphoid Cells"

_nutrients, 2019, doi:10.3390/nu11122998_

Round 1
Reviewer 1 Report
In their manuscript Kobbayahi et al demonstrate that the oral application of yogurt fermenting S. thermophilus and L. bulgaricus is inducing the Reg3 family of anti-microbial peptides in the small intestinum of mice. They try to attribute this to the induction of cytokines IL-23 and IL-22 in lamina propria dendritic cells and ILC-3 cells, respectively. These are interesting results but some points should be clarified before publication.
1. The English and style should be improve. Sometimes it is difficult to extract the correct meaning of some sentences.
2. In general: it is not clear whether the data were reproduced. How often were the experiments repeated?
3. The authors compare always heat-killed bacteria in vitro with life bacteria in vivo. Why is that? Living bacteria behave very much different from hk bacteria.
4. The legend of the figures should be adjusted.
5. I am surprised by the statistics of Reg-3 measurement. 2038 should be not significant compared to 1131.
6. Simlarly, stats of 3b are surprising. Anyway the staining is not convincing. What would happen if other bacteria would be applied?
7. Figure 5. Living bactria should have been tested.
8. No direct connectivity between the induction of IL-23 and IL-22 and the induction of Reg-3 has been established. This should be discussed.
Reviewer 2 Report
Review of Manuscript ID: Nutrients-637356
Type of manuscript: Article
Title: Lactobacillus delbrueckii subsp. bulgaricus 2038 and Streptococcus thermophilus 1131 induce the expression of REG3 family in the small intestine of mice via stimulation of dendritic cells and type 3 innate lymphoid cells
Researchers of the manuscript titled “Lactobacillus delbrueckii subsp. bulgaricus 2038 and Streptococcus thermophilus 1131 induce the expression of REG3 family in the small intestine of mice via stimulation of dendritic cells and type 3 innate lymphoid cells” reported that Lactobacillus delbrueckii 17 subsp. bulgaricus 2038 and Streptococcus thermophilus 1131 contained in fermented yogurt are able to induce the expression of REG3G gene via stimulation of IL23 protein production.
The REG3G (Regenerating Family Member 3 Gamma) gene encodes a regIIIg (regenerating islet-derived protein III-gamma also regenerating islet-derived protein 3 gamma; reg3g) protein.
The work is written well, but there are many shortcomings in writing.
Line 18 -19 “induced the gene expression of REG3 family, a kind of antimicrobial peptides in the small intestine, “ Is this true?
Line 29-31 ” Our findings suggested that these yogurt starter strains, L. bulgaricus 2038 and S. thermophilus 1131, have the potential to maintain and improve the intestinal barrier function by stimulating immune cells in the LP.” Is this true?
There are no common explanations for the abbreviations, sometimes I observe their lack. The same goes for materials and methods, where there is lack town and country. The name CFU (Colony-Forming Units) is not explain in the text, moreover, it was accepted that we write it in capital letters.
I tried to mark it in the manuscript in the form of comments.
Genes are written in italics - see HUGO Gene Nomenclature Committee.
Figures 3a and 4a are not clear, you can improve graphical side of them.
In figure 5 you write "relative activity of IL23" and in the text "relative to these strain". Can you explain how it was calculated?
Please check attached manuscript in pdf format with comments.

Round 2
Reviewer 1 Report
Although I dislike that several experiments had been done only once and not been reproduced, the paper is ready for publication. For the future the authors should be advised to at least reproduce result once.
Reviewer 2 Report
Reply to Authors replies:
Thank you for your careful correction. You did a great job. I accepted.
But if going on gene, I think that you do not understand me.
I suggested completing the line 19 …gene expression of the REG3 (regenerating family member3) family, which encode a kind of antimicrobial .…
The REG gene family consists of the gene members, which encode proteins having antimicrobial properties.
The same in line 59
If you consider it appropriate to complete the sentences shown, the work will be without mental shortcuts that are committed in many works (sometimes genes are not distinguished from proteins, which is a mistake when you are estimated genes).
